# An Estimation of the Leaf Nitrogen Content of Apple Tree Canopies Based on Multispectral Unmanned Aerial Vehicle Imagery and Machine Learning Methods

Xin Zhao [1,2], Zeyi Zhao [1,2], Fengnian Zhao [1,2], Jiangfan Liu [1,2], Zhaoyang Li [1,2], Xingpeng Wang [1,2,3,*] and Yang Gao [3,4,*]

1   College of Water Resource and Architecture Engineering, Tarim University, Aral 843300, China; 15730999979@163.com (X.Z.); zeyizhao@aliyun.com (Z.Z.); zfn19990411@163.com (F.Z.); jiangfangliu@aliyun.com (J.L.); lizhaoyang2i1@163.com (Z.L.)
2   Key Laboratory of Northwest Oasis Water-Saving Agriculture, Ministry of Agriculture and Rural Affairs, Shihezi 832000, China
3   Institute of Western Agriculture, Chinese Academy of Agricultural Sciences, Changji 831100, China
4   Institute of Farmland Irrigation, Chinese Academy of Agricultural Sciences, Xinxiang 453000, China
*   Correspondence: 13999068354@163.com (X.W.); gaoyang@caas.cn (Y.G.)

**Abstract:** Accurate nitrogen fertilizer management determines the yield and quality of fruit trees, but there is a lack of multispectral UAV-based nitrogen fertilizer monitoring technology for orchards. Therefore, in this study, a field experiment was conducted by UAV to acquire multispectral images of an apple orchard with dwarf stocks and dense planting in southern Xinjiang and to estimate the nitrogen content of canopy leaves of apple trees by using three machine learning methods. The three inversion methods were partial least squares regression (PLSR), ridge regression (RR), and random forest regression (RFR). The results showed that the RF model could significantly improve the accuracy of estimating the leaf nitrogen content of the apple tree canopy, and the validation set of the four periods of apple trees ranged from 0.670 to 0.797 for $R^2$, 0.838 mg $L^{-1}$ to 4.403 mg $L^{-1}$ for RMSE, and 1.74 to 2.222 for RPD, among which the RF model of the pre-fruit expansion stage of the 2023 season had the highest accuracy. This paper shows that the apple tree leaf nitrogen content estimation model based on multispectral UAV images constructed by using the RF machine learning method can timely and accurately diagnose the growth condition of apple trees, provide technical support for precise nitrogen fertilizer management in orchards, and provide a certain scientific basis for tree crop growth.

**Keywords:** drone multispectral; machine learning; remote sensing inversion; apple tree

## 1. Introduction

Nitrogen is the basic constituent of protein, chlorophyll, amino acids, and other key organic molecules [1] and is a key indicator of plant growth and also the main concern of precision agriculture. Nitrogen deficiency dictates the synthesis of other substances, such as chlorophyll and amino acids, thereby reducing the photosynthetic capacity of plants [2], which in turn affects crop growth, yield, and quality, posing a risk to food security [3]. However, the overuse of nitrogen is harmful to the environment [4], mainly due to the evaporation of nitrogen in the atmosphere and leaching into the groundwater, resulting in water and atmospheric pollution as well as the risk of greenhouse gas (GHG) emissions; but globally, especially in Europe, nitrogen overuse has slowly become less threatening. Therefore, there is a growing need for precise nitrogen fertilizer management strategies [5,6].

The evaluation of plant nitrogen status is mainly focused on estimating leaf nitrogen content [7–9]. Therefore, researchers usually use differences in plant varieties and

---

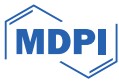

growth stages as the basis for diagnosing plant nitrogen nutrition. Traditional nitrogen determination requires destructive sampling and has a time lag. With the development of remote sensing technology, the monitoring of plant nitrogen has new technical support. Ground-based remote sensing has the irreplaceable advantages of being free from sunlight, soil, and weeds [10], such as when using Multiplex[®]3 (Dynamax, Elkhart, IN, USA) [11], Dualex 4 (Force-A, Orsay, Paris, France) [12–14], and other proximal remote sensing sensors, and has produced excellent results. However, it still requires a lot of repetitive work at the test site and only obtains spectral information from a single location, making it difficult to realize nitrogen content monitoring on a large regional scale. Satellite remote sensing platforms can realize nitrogen content monitoring on a regional scale, and there have been several results produced using this method [15,16], but the resolution of satellite remote sensing is low, and satellite images have a fixed transit date, so the flexibility of obtaining satellite remote sensing images is not high. Some scholars also use methods such as remote sensing, topographic maps, and geographic information system (GIS) analysis to estimate tree crown characteristics. Valjarević et al. [17] used these methods to reconstruct forest conditions in the Toplica region over a period of nearly 60 years. Zhou et al. [18] investigated the differences in park access by race/ethnicity in six medium-sized Illinois cities, and differences in canopy cover were also examined using a variety of classification techniques to calculate the number of canopies in a community.

In recent years, UAV remote sensing has gradually become the mainstream method of digital precision agriculture in the field of remote sensing in agriculture due to some advantages such as its small size, flexibility, portability, low cost, etc., and has been successfully applied to the prediction of crop chlorophyll [19] and crop yield [20]. The camera carried by the UAV usually acquires visible, multispectral, and hyperspectral images, and their main difference lies in the differences in the number of bands, resolution processing methods, etc. The spectral information of UAV remote sensing images is extracted to calculate the vegetation index, and the screening of sensitive variables can effectively remove the redundant information in the vegetation index, reduce the complexity of the model inversion, and thus improve the model accuracy. For example, Noguera et al. [21] extracted the multispectral UAV information of the olive tree canopy and constructed a prediction model for nitrogen, phosphorus, and potassium content of the olive tree canopy. Prado Osco et al. [22] extracted the spectral information of the multispectral images combined with a machine learning method to predict the nitrogen concentration of the citrus tree canopy. The same research method was sampled to estimate the nitrogen concentration of the crop in different tree crops, and better research results were achieved [23,24]. However, few of the available research crops have predicted canopy leaf N content for apple trees.

Apple occupies an important position in China's fruit production, and its planting area and production are located at the forefront of global food production. Due to the unique geographical environmental and climatic characteristics, the Xinjiang production area is the only independent production area among the six major apple production areas in the country, and the output accounts for about 4.5% of the national total output, while South Xinjiang is the main production area in Xinjiang. Plant growth monitoring plays an important role in apple yield and quality enhancement. In southern Xinjiang, orchards have large areas and a low level of precision management of irrigation and fertilizer application. Traditional methods for measuring the water and nutrient status of orchards require a large amount of fieldwork, which is undoubtedly time-consuming and labor-intensive [25]. The development of UAV remote sensing provides a feasible way to accurately monitor the plant growth of orchards. Caruso et al. [26] used multispectral UAV map images and thermograms to monitor olive tree water status, canopy growth, and yield and concluded that there are different effects on fruit tree growth and yield under different irrigation conditions, etc. Apolo et al. [27] estimated the yield and size of citrus trees using a stick method combining UAVs and deep learning. Arakawa et al. [28] estimated the yield and size of citrus trees using a machine learning method combining UAV and UAV visible light images to estimate the yield of chestnut fruits on trees. Zhang et al. [29] used ground-space

remote sensing fusion to invert the nitrogen content of orchard canopies. Fruit trees are perennials that grow and develop differently from annual crops, and the complexity of their canopies still leaves gaps in the monitoring of canopy moisture and nutrient status.

Therefore, the objective of this paper was to invert the nitrogen content of the apple tree canopy of an orchard with dwarf stocks and dense planting in southern Xinjiang using three different methods based on spectral data imaged by an unmanned aerial vehicle (UAV). The methods used include partial least squares regression (PLSR), mountain ridge regression (RR), and random forest regression (RF).

## 2. Materials and Methods

### 2.1. Study Area

The field experiment was carried out in an apple orchard (40°39′ N, 81°16′ E, average elevation 1013 m) with dwarf stocks and dense planting located in Alar City (Figure 1), Xinjiang, China. The experimental region has a typical arid climate. Changes in mean annual temperature and rainfall are shown in Figure 2, while the annual evaporation is as high as about 2100 mm, and the average annual total solar radiation is 552.73 kJ cm$^2$, with an average annual sunshine duration of 2900 h, a frost-free period of 203 d, and a fluctuating groundwater level of less than 3.0 m. The area of experimental plots is 2100 m$^2$, the soil texture is sandy loam, the field water capacity of 0–120 cm layer is 18.5%, the soil density is 1.51 g cm$^{-3}$, and the basic physical and chemical properties of the soil are shown in Table 1.

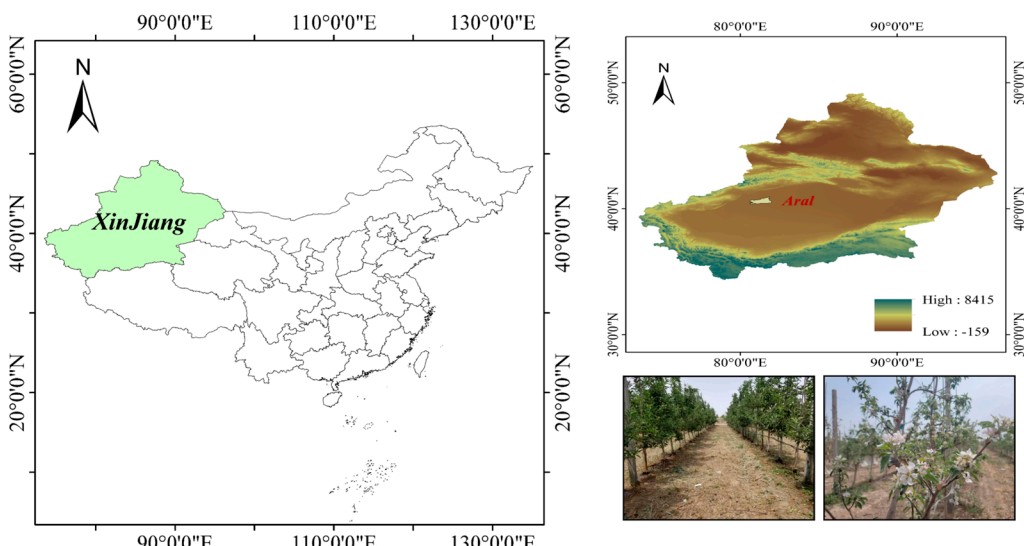

**Figure 1.** Overview map of the apple orchard with dwarf stocks and dense planting.

**Table 1.** Basic soil physical and chemical properties in the experimental site.

| Organic Matter (g kg$^{-1}$) | Available Phosphorus Content (mg kg$^{-1}$) | Available Boron Content (mg kg$^{-1}$) | Available Potassium Content (mg kg$^{-1}$) | Alkali Hydrolyzed Nitrogen Content (mg kg$^{-1}$) |
|---|---|---|---|---|
| 11.05 | 7.2 | 0.6 | 33 | 18.4 |

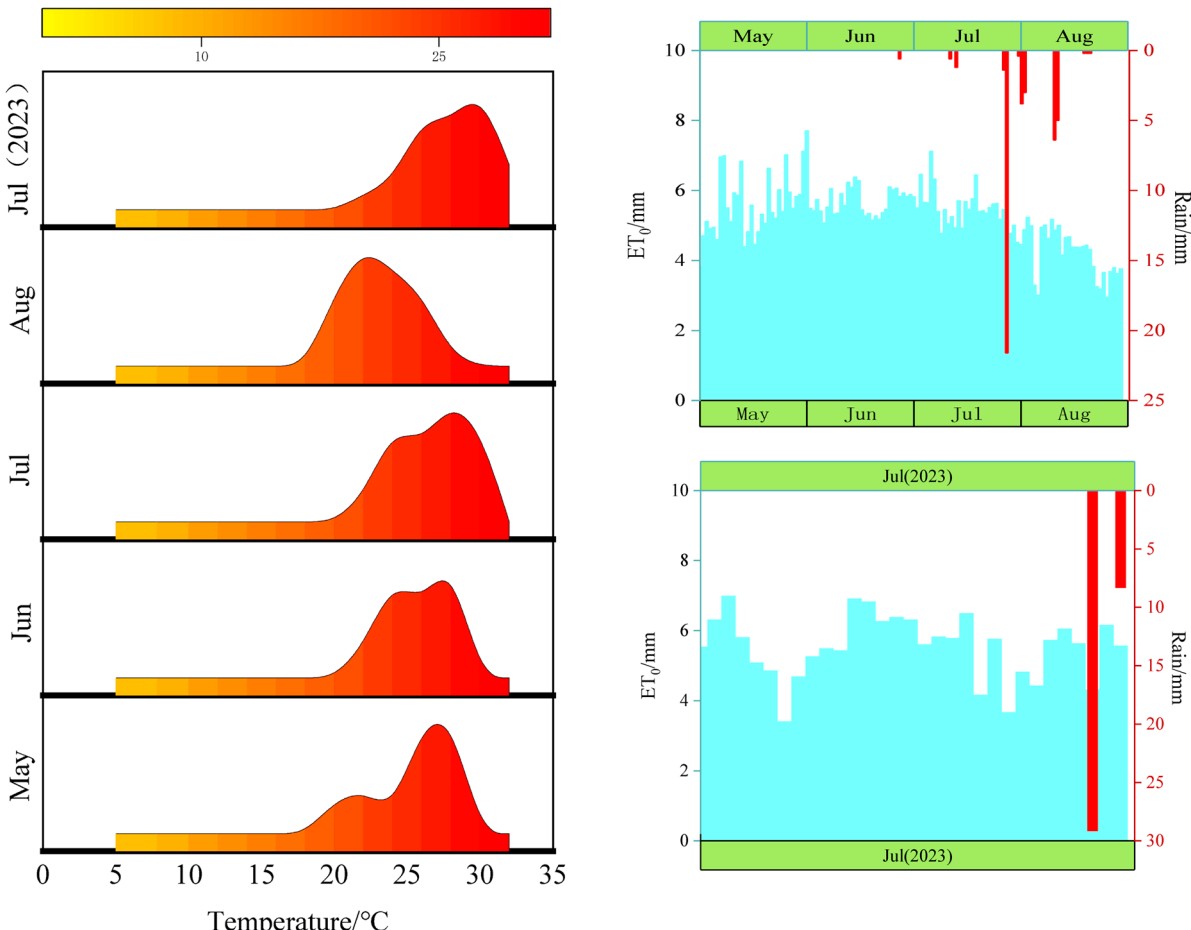

**Figure 2.** Average fruit tree fertility and rainfall.

### 2.2. Experimental Design

The variety was 5-year-old 'Royal Gala' with row spacing of 3.5 m × 1 m and plant height of about 4.5 m. The plant was grown in the same row. Five irrigation ratings of 13.5 mm ($W_1$), 18 mm ($W_2$), 22.5 mm ($W_3$), 27 mm ($W_4$), and 31.5 mm ($W_5$) were set up, and the irrigation regime was as shown in Table 2. Figure 2 shows the temperature and rainfall during the experimental period of 2022 and 2023. The drip irrigation method was one row and one pipe, with a drip head flow rate of 4 L/h, a drip hole spacing of 30 cm, and drip pipes arranged on bamboo poles 50 cm above the ground.

**Table 2.** Table of irrigation regimes.

| Vintages | Treatment | Flooding Quota | Number of Waterings/Times | Irrigation Quota/mm |
|---|---|---|---|---|
| | $W_1$ | 13.50 | 21 | 283.50 |
| | $W_2$ | 18.00 | 21 | 378.00 |
| 2022 | $W_3$ | 22.50 | 21 | 472.50 |
| | $W_4$ | 27.00 | 21 | 567.00 |
| | $W_5$ | 31.50 | 21 | 661.50 |
| | $W_1$ | 13.50 | 21 | 283.50 |
| | $W_2$ | 18.00 | 21 | 378.00 |
| 2023 | $W_3$ | 22.50 | 21 | 472.50 |
| | $W_4$ | 27.00 | 21 | 567.00 |
| | $W_5$ | 31.50 | 21 | 661.50 |

### 2.3. UAV Imagery of Apple Tree Canopy

A DJI M600 UAV equipped with a Micro-MCA Snap multispectral imager (Tetracam Inc., Chatsworth, CA, USA) was used to acquire cotton canopy images, as shown in Figure 3. The Micro-MCA Snap multispectral camera captures bands from 450 nm to 1000 nm, with a sensor of 1.3 million pixels and a lens focal length of 9.6 mm, and the sensor parameters are listed in Table 3. The maximum flying altitude of the aircraft is 2500 m, the maximum flying speed is 18 m/s, and the hovering accuracy is 0.5 m vertically and 1.5 m horizontally.

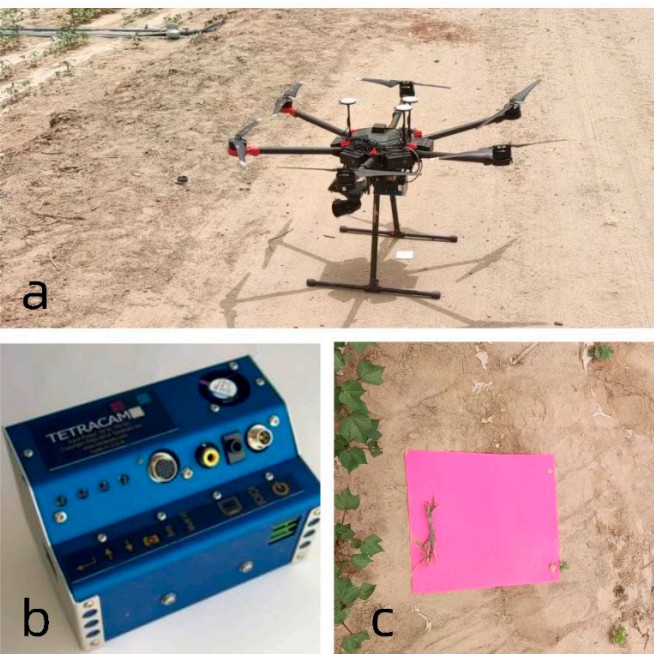

**Figure 3.** (**a**) aerial drone; (**b**) Micro-MCA Snap multispectral camera; (**c**) ground identification mark.

**Table 3.** Multispectral camera sensor parameters.

| Band Name | Center Wavelength (nm) | Band Width (nm) |
| --- | --- | --- |
| NIR1 (near-infrared 1 band) | 800 | 80 |
| B (blue light band) | 490 | 80 |
| G (green light band) | 550 | 70 |
| R (red light band) | 680 | 80 |
| RE (red side band) | 720 | 100 |
| NIR2 (near-infrared 2 band) | 900 | 140 |

### 2.4. Multispectral Imaging and Spectral Data Acquisition

#### 2.4.1. Multispectral Image

Multispectral imaging took place on 1 May (flowering and fruiting period), 4 July (pre-expansion period), and 2 August (post-expansion period) in 2022 season and on 12 July (pre-expansion period) in 2023 season. Each flight was flown between 12:00 and 16:00 when the sky was clear and less cloudy and when the field of view was wide and suitable for UAV flight. The drone was set to fly at an altitude of 50 m [30], with a set airspeed of 7 ms$^{-1}$ and an 85% overlap between the heading and the side image. The Micro-MCA Snap multispectral camera has a focal length of 9.6 mm, lens shot 90 degrees vertically down, an image size of 1280 × 1024, and a ground resolution of approximately 4 cm. Ground identification markers were laid down before the aircraft flew, and the sampling points were fixed for each flight, and the images taken by the UAV contained the entire test area, which could be clearly labeled in the images. The sampled spectral information can be extracted using ArcGIS.

### 2.4.2. Spectral Data Acquisition and Processing

For the multispectral images of the apple trees taken by the UAV in three periods, the pix4dmapper software (pix4, Lausanne, Switzerland) was used to stitch together the successive photographs to obtain multispectral images in six separate bands. Band synthesis was then performed using ENVI5.6 software (EVIS, Charlottesville, VA, USA) to obtain a six-band multispectral image, and the file was exported to TIF format for post-processing. Next, the spectral image was cropped using ArcGIS 10.8 software (ESRI, Redlands, CA, USA) to obtain the target region. ArcGIS 10.8 software was used to extract the pixel reflectance of each sampling point.

### 2.5. Total Nitrogen Content of Apple Leaves

Total nitrogen content was determined by the Kjeldahl method [31]. The day after the drone flew at each fertility stage, the leaves were collected, 60 sampling points were randomly selected in the experimental area, and 90 sampling sites were selected in the pilot area in 2023. The leaves were picked uniformly in each plot. To ensure no loss of nitrogen content in apple leaves, the leaves were dried at 60 °C for one week, and the dried leaves were ground and sieved through a 0.2 mm sieve. A total of 0.2 g of leaf sample was heated in 5 mL of concentrated sulfuric acid at a concentration of more than 95%, and catalytic boiling was carried out with hydrogen peroxide at a concentration of 30%. The decoction solution was subjected to an AA3 Continuous Flow Analyzer (SEAL, Norderstedt, Germany) to test the nitrogen content of the samples.

### 2.6. Determination of Vegetation Index

Vegetation indices formed by combining reflectance of different bands can eliminate the interference of external factors and improve the sensitivity of target parameters. In this paper, 14 vegetation indices (Table 4), which are closely related to the nitrogen status of fruit trees, were selected to establish a prediction model for the nitrogen content of apple tree canopy leaves at different fertility periods.

**Table 4.** Calculation of the vegetation index.

| Vegetation Index | Formula | References |
|---|---|---|
| VARI | $(G - R)/(G + R - B)$ | [32] |
| NRI | $(G - R)/(G + R)$ | [33] |
| SAVI | $(1 + 0.5) \times (NIR - R)/(NIR + R + 0.5)$ | [34] |
| EVI | $2.5 \times (NIR - R)/(NIR + 6 \times R - 7.5 \times B + 1)$ | [35] |
| R-M | $NIR/RE - 1$ | [36] |
| NDRE | $(NIR - RE)/(NIR + RE)$ | [37] |
| GOSAVI | $1.16 \times (NIR - G)/(NIR + G + 0.16)$ | [38] |
| OSAVI | $1.16 \times (NIR - R)/(NIR + R + 0.16)$ | [39] |
| GBNDVI | $NIR - G + B/NIR + G + B$ | [40] |
| NDVI | $(NIR - R)/(NIR + R)$ | [41] |
| RVI | $NIR/R$ | [42] |
| DVI | $NIR - R$ | [43] |
| GNDVI | $NIR - G/NIR + G$ | [40] |
| TVI | $0.5 \times (120 \times (NIR - G) - 200 \times (R - G)$ | [44] |

Note: R is the red band, G is the green band, NIR is the near-red band, and RE is the red edge band.

### 2.7. Machine Learning Models

In this paper, we used Python learning platform to divide the modeling set and validation set in the ratio of 7:3, with leaf nitrogen content of fruit trees at each reproductive stage as the dependent variable and vegetation index as the independent variable. Firstly, the nitrogen content estimation models of each fertility period were established to explore the optimal spectral indices, and secondly, partial least squares regression (PLSR), ridge regression (RE), support vector machine regression (SVR), and random forest regression

(RFR) models were constructed. The accuracy of the canopy nitrogen content of apple trees in each model was estimated to select the optimal model.

PLSR [45] is an innovative bilinear regression method for evaluating multivariate statistics. RR [46] is a biased estimation regression method dedicated to the analysis of covariate data and is a modified partial least squares estimator, but it is superior to the partial least squares method in fitting certain data. RFR [47], proposed by Breiman in 2001, generates a training set by integrating the decision tree and averaging the predictions after continuous regression and multiple sampling of the samples.

*2.8. Model Evaluation Analysis*

In this paper, a model for estimating the nitrogen content of fruit tree canopy leaves was evaluated using the coefficient of determination ($R^2$), root mean square error (RMSE), and relative prediction deviation (RPD). The calculation formulae are given in Equations (1)–(3).

$$R^2 = 1 - \frac{\sum_{i=1}^{n} (fi - y)^2}{\sum_{i=1}^{n} (fi - \overline{y})^2} \tag{1}$$

$$RMSE = \sqrt{\frac{\sum_{i=1}^{n} (fi - y)}{n}} \tag{2}$$

$$RPD = \frac{SD_{fi}}{RMSE} \tag{3}$$

where $n$ is the number of samples, $fi$ is the measured value, $y$ is the predicted value, $SD_{fi}$ is the standard deviation, and $\overline{y}$ is the average of the measured values.

The correlation between nitrogen content and each vegetation index was determined using the Pearson correlation coefficient. The correlation coefficient is generally expressed as $r$, with a range of $[-1, 1]$. When $r = 0$, it indicates that there is no linear relationship between the two variables; when $r$ is less than 0, it indicates a negative correlation; and when $r$ is greater than 0, it indicates a positive correlation. When the absolute value of $r$ is in (0, 0.2), there is a very weak correlation; when it is (0.2, 0.4), there is a weak correlation; when it is (0.4, 0.6), there is a medium correlation; when it is (0.6, 0.8), there is a strong correlation; and when it is (0.8, 1), there is a very strong correlation. In this paper, vegetation indices with absolute values of correlation greater than 0.4, i.e., above moderate correlation, are selected for modeling. The calculation formula is shown in Equation (4).

$$r = \frac{\left[\sum_{i=1}^{n} (xi - \overline{x})(yi - \overline{y})\right]^2}{\sum_{i=1}^{n} (xi - \overline{x})^2 \sum_{i=1}^{n} (yi - \overline{y})^2} \tag{4}$$

where $xi$ and $yi$ are the values of the two variables, and $\overline{x}$ and $\overline{y}$ are the mean values of the variables.

The closer the $R^2$ is to 1, the smaller the RMSE, the better the model, and the RPD is the ratio of the standard deviation (SD) to the root mean square error (RMSE) of the test set, which indicates that the model has no predictive ability when the RPD is <1.4. During the construction of the model for estimating nitrogen concentration in apple tree canopy leaves, a network search was used to determine the super parameters, and the optimal parameters of each model were identified by h (h is an arbitrary number) times the cross-validation comparisons.

The technical roadmap of this study is shown in Figure 4.

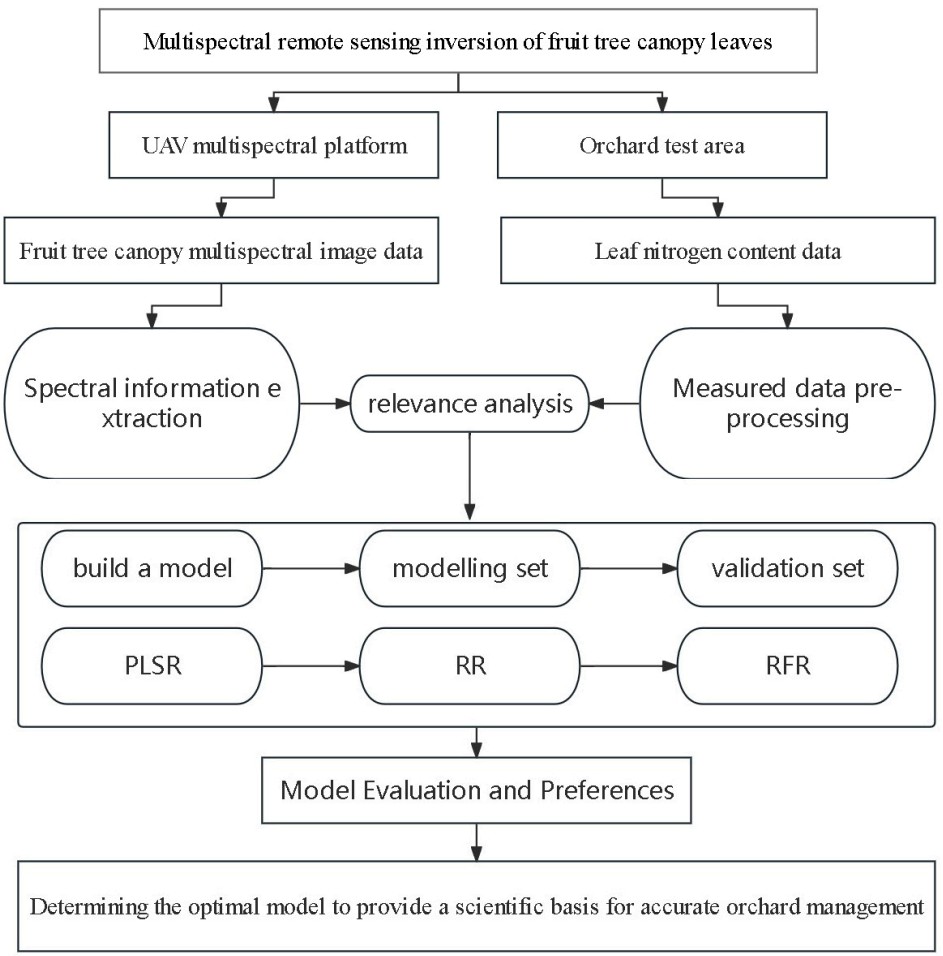

**Figure 4.** Technology roadmap.

## 3. Results

### 3.1. Relationships between the Nitrogen Content of the Canopy Leaves and Vegetation Index

The statistical description of the nitrogen content of the canopy leaves of apple trees in each reproductive period is shown in Table 5. Due to the dilution effect, the nitrogen content of the apple tree canopy leaves decreased as fertility increased.

**Table 5.** Statistical description of the nitrogen content of apple tree canopy leaves.

| Growth Phase | Number of Samples | Maximum Value (mg/L) | Minimum Value (mg/L) | Average Value (mg/L) | Standard Deviation | CV (%) |
|---|---|---|---|---|---|---|
| Whole | 180 | 57.25 | 16.52 | 28.42 | 7.96 | 28.0 |
| Flowering | 60 | 57.25 | 23.02 | 36.66 | 7.27 | 19.8 |
| Preliminary | 60 | 34.96 | 21.12 | 27.57 | 2.95 | 10.7 |
| Expansion | 60 | 27.62 | 16.52 | 21.04 | 2.27 | 10.8 |
| Preliminary2 | 80 | 43.94 | 22.66 | 33.31 | 4.66 | 14 |

In this study, Pearson's phase relationship was used to analyze the correlation between canopy leaf nitrogen content and the vegetation index of fruit trees. Origin was used to produce correlation heatmaps, and the numbers indicate the correlation between them; when the correlation is higher, the corresponding pattern is larger and darker. Figures 5–7 show the correlation between the leaf nitrogen content of fruit trees and vegetation indices in 2022 for each reproductive stage. The results show that among the 14 vegetation indices selected, the leaf nitrogen content of fruit trees in each reproductive stage was positively correlated with vegetation indices, and the correlation was concentrated in the range

between 0.4 and 0.7. Figure 8 shows the correlation between the N content of fruit trees and vegetation indices during the pre-fruit expansion period in 2023, and the results showed that all vegetation indices except DVI were positively correlated with N content.

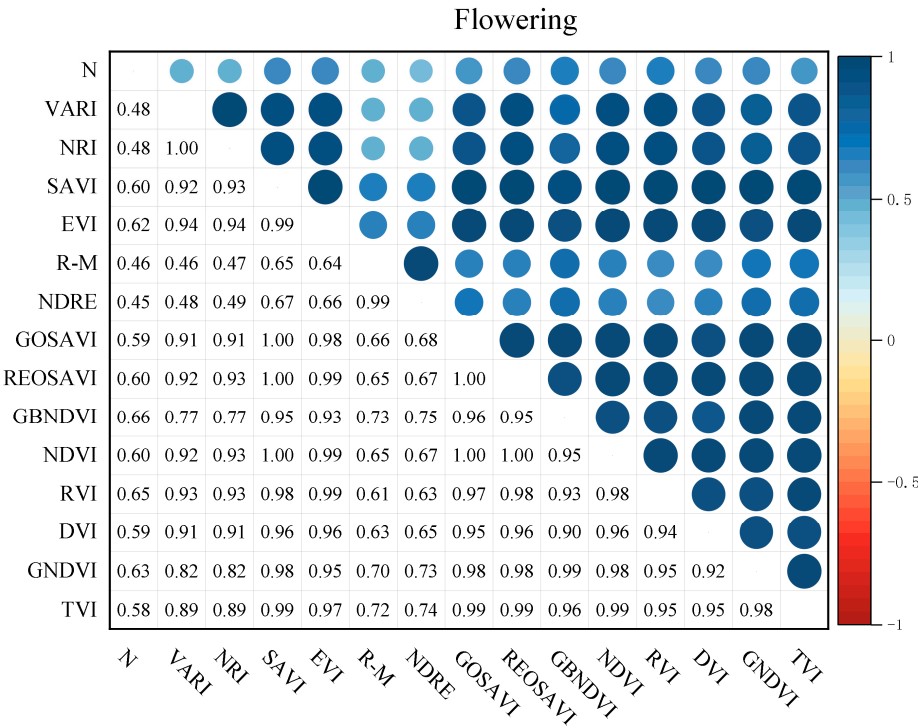

**Figure 5.** Heat map of nitrogen content during flowering and fruiting in relation to vegetation index.

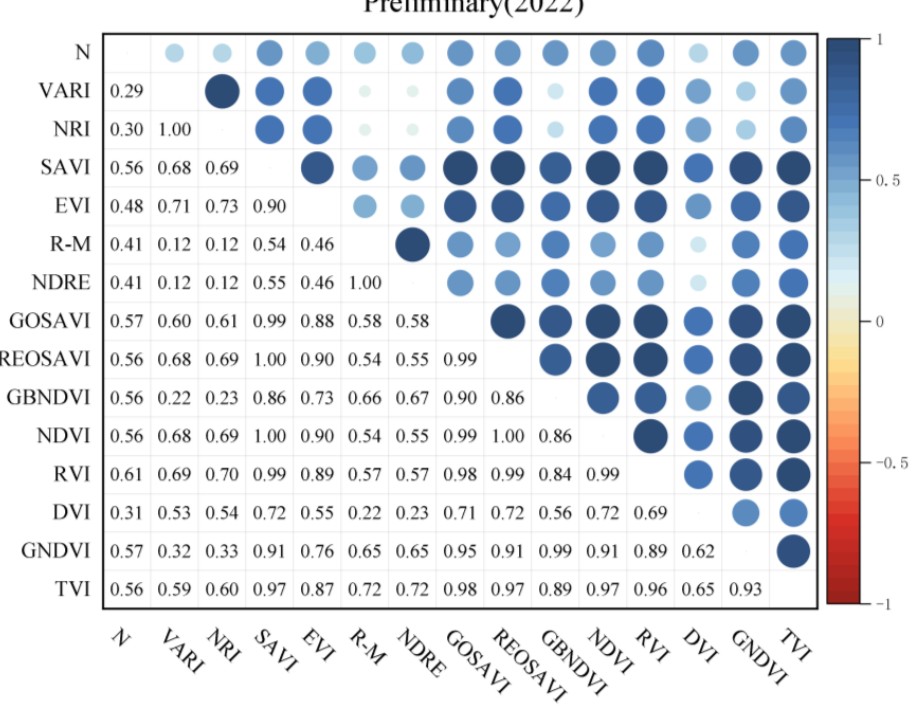

**Figure 6.** Heat map relating nitrogen content to vegetation index during pre-fruit expansion in 2022.

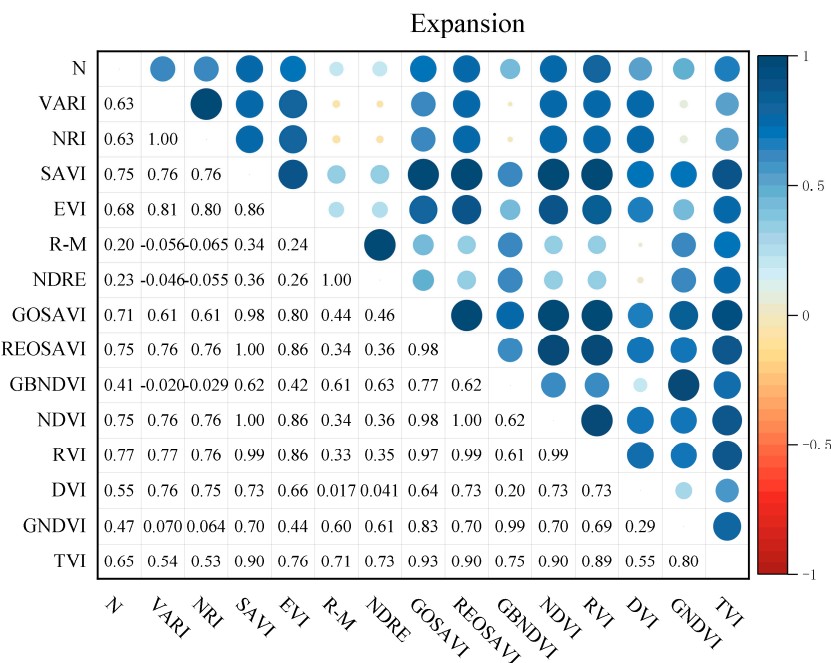

**Figure 7.** Heatmap of nitrogen content in relation to vegetation index at late fruit expansion stage.

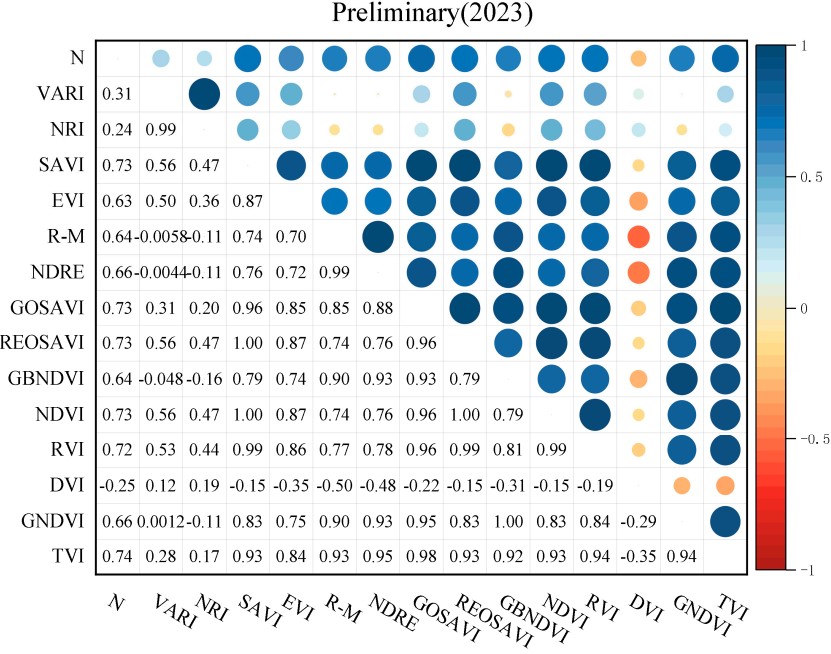

**Figure 8.** Heat map relating nitrogen content to vegetation index during pre-fruit expansion in 2023.

### 3.2. Machine Learning Model Analysis for Estimating Nitrogen Content of Apple Leaves

Figure 9 shows the estimated model for estimating the canopy leaf nitrogen content of fruit trees at each fertility stage using the PLSR model. During the three fertility periods of fruit trees in 2022, pre-fruit expansion and post-fruit expansion, the validation set $R^2$ varied in the range of 0.574 to 0.575, and the RMSE varied in the range of 1.904 mg·$L^{-1}$ to 1.908 mg·$L^{-1}$, indicating that the accuracy of estimation using the PLSR model was similar during these two fertility periods. And at flowering and fruiting, it had a validation set of $R^2 = 0.690$ and RMSE = 3.942 mg·$L^{-1}$, and at pre-fruit expansion in 2023, it had a validation set of $R^2 = 0.650$ and RMSE = 2.780 mg·$L^{-1}$. As far as the modeling set is concerned, the best fertility period for predicting the leaf nitrogen content of fruit trees using the PLSR model was the late fruit expansion stage in 2022. It had the highest modeling set $R^2$ and

the smallest RMSE among all the fertility periods, with $R^2 = 0.729$, RMSE = 1.001 mg·L$^{-1}$, and RPD = 1.922, which had a certain prediction effect.

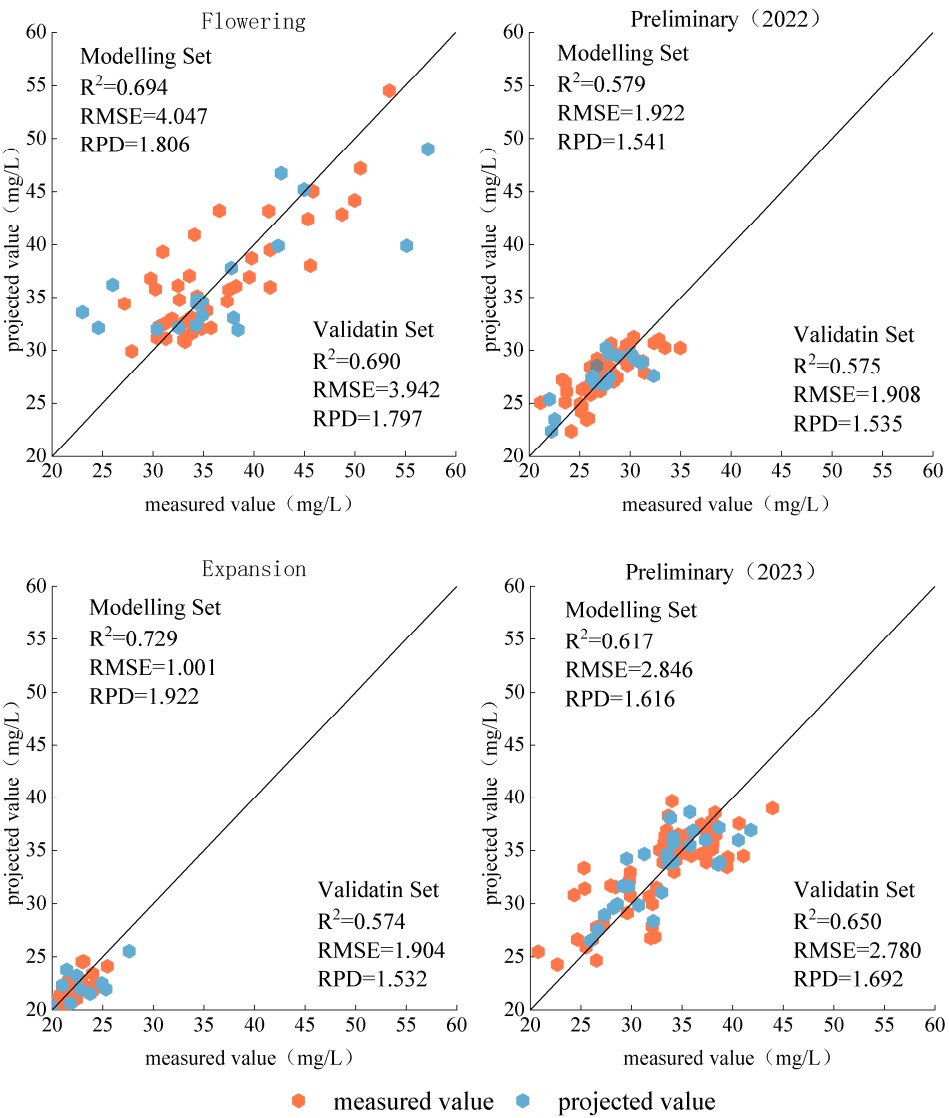

**Figure 9.** Distribution of PLSR model measured and predicted values. The 1:1 line is in greyscale.

Figure 10 shows the estimated model for estimating canopy foliar N content of fruit trees at various fertility stages using the RR model. The validation set $R^2$ varied between 0.480 and 0.628, and the RMSE ranged from 1.134 to 3.651 during the three fertility periods of the fruit trees, 2022, pre-fruit expansion and post-fruit expansion, whereas the validation set for the 2023 pre-fruit expansion was $R^2 = 0.559$ and RMSE = 2.568 mg·L$^{-1}$. As far as the modeling set is concerned, the best fertility period for predicting the leaf nitrogen content of fruit trees using the RR model was the late fruit expansion stage in 2022. It had the highest modeling set $R^2$ and the lowest RMSE among all the fertility periods with $R^2 = 0.707$, RMSE = 1.079 mg·L$^{-1}$, and RPD = 1.847.

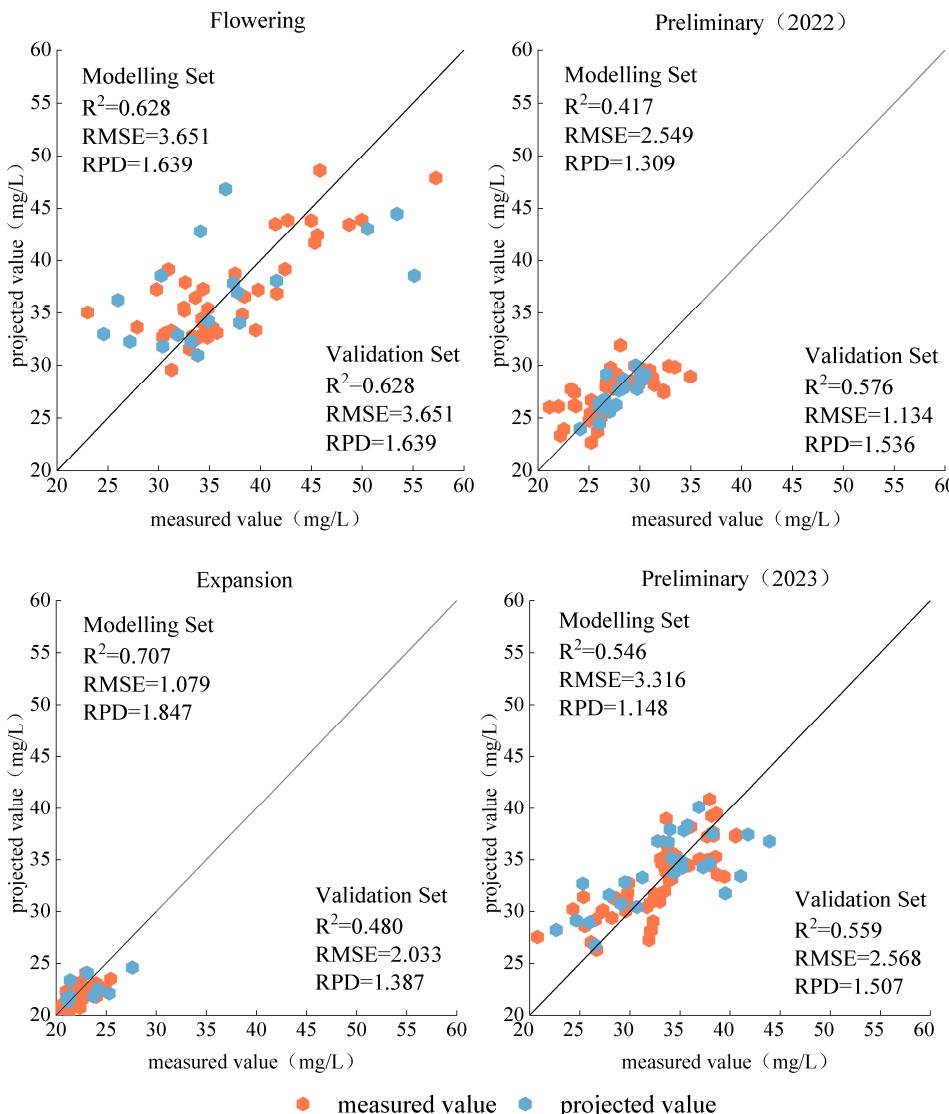

**Figure 10.** Distribution of measured and predicted values for the RR model. The 1:1 line is shown in greyscale.

The validation results of the RF model in predicting the nitrogen content of fruit tree leaves at all fertility stages are presented in Figure 11. Its validation set $R^2$ varied from 0.570 to 0.758, and RMSE varied from 0.838 mg·L$^{-1}$ to 4.403 mg·L$^{-1}$ during the three fertility periods of the fruit trees in 2022, and its validation accuracy became higher, and RMSE decreased as the fertility period progressed, indicating that the accuracy of the model improved. It also had a good validation set of $R^2$ = 0.797, RMSE = 1.891 mg·L$^{-1}$, and RPD = 2.222 in 2023 during the pre-fruit expansion period.

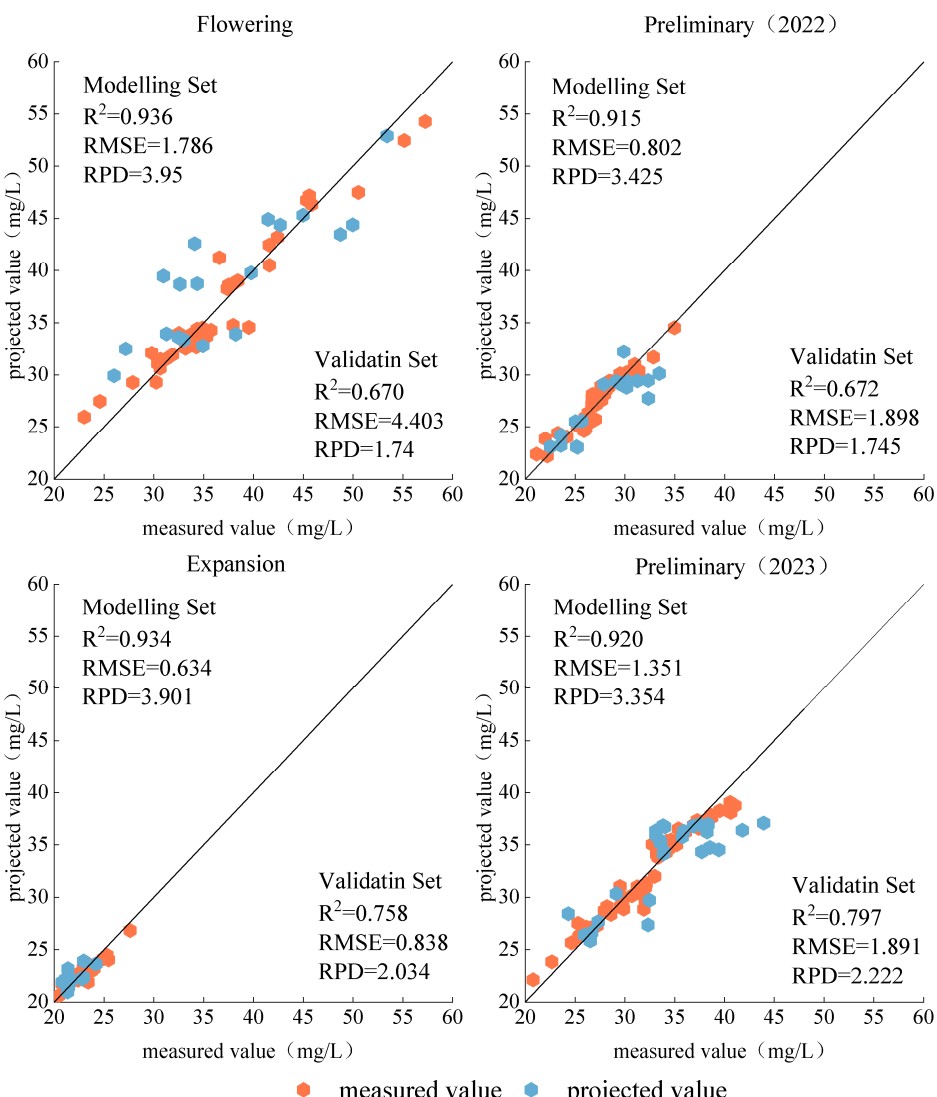

**Figure 11.** Distribution of measured and predicted values for the RF model. The 1:1 line is shown in greyscale.

### 3.3. Comparison of Model Accuracy

The RPD of the validation set was further utilized to assess the accuracy of the estimation of canopy leaf nitrogen content of fruit trees at various fertility stages, as shown in Figure 12. At the flowering and fruiting stage in 2022, the highest accuracy was predicted using the PLSR model, which had a predictive effect of RPD = 1.797 (>1.4), followed by the RF model, which had a predictive effect of RPD = 1.74 (>1.4), and lastly, the RR model, which had a predictive effect of RPD = 1.639 (>1.4). In the pre-expansion period of 2022, the highest accuracy was obtained by using the RF model, which had an RPD = 1.745 (>1.4), followed by the RR model and the PLSR model, which had an RPD = 1.536 (>1.4) and 1.535 (>1.4), respectively, indicating that both of them had similar prediction effects. At the late fruit expansion stage in 2022, the highest accuracy was obtained using the RF model with an RPD = 2.034 (>1.4), followed by the PLSR model with an RPD = 1.532 (>1.4), and lastly, the RR model with an RPD = 1.387 (<1.4), indicating that the model was not effective in prediction at this fertility stage. In 2023, the best prediction in the pre-expansion stage of the fruit was made using the RF model, which had an RPD = 2.222 (>1.4), followed by the PLSR model, which had an RPD = 1.692 (>1.4), and lastly, the RR model, which had an RPD = 1.507 (>1.4), which was effective in making a prediction. Overall, the RF model was

used to predict the canopy leaf nitrogen content of fruit trees with good predictive effect at all fertility stages.

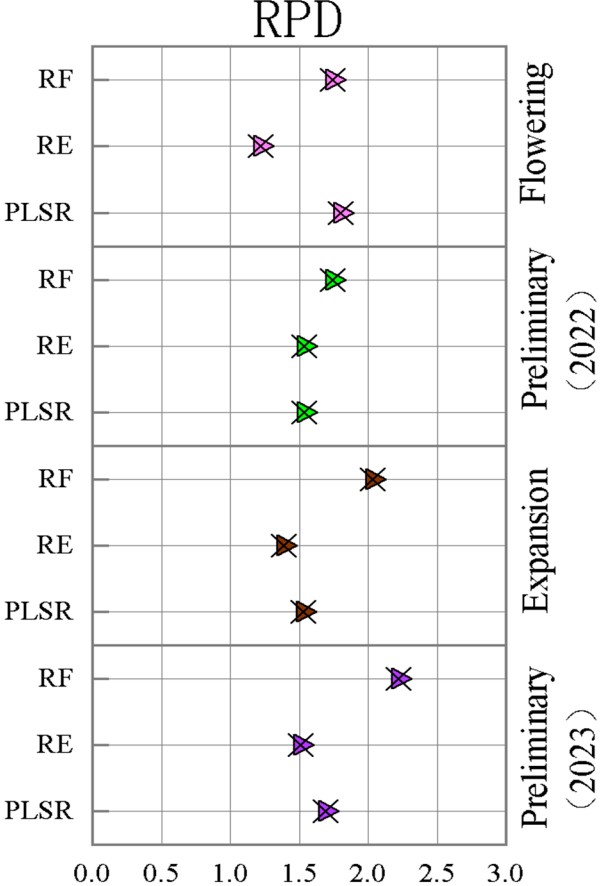

**Figure 12.** Comparison of RPD values among models for estimating leaf nitrogen content in the canopy of apple trees.

## 4. Discussion

### 4.1. Estimation of Nitrogen Content Using Vegetation Index as the Independent Variable

Due to the different absorption and scattering effects of incident light in different wavelength bands, vegetation indices are able to develop specific spectral response characteristics [48]. In this study, Pearson correlation analysis was used to obtain the correlation between 14 vegetation indices and the nitrogen content of fruit tree leaves at various fertility stages. The results showed that the correlations between vegetation indices and nitrogen content were inconsistent at different fertility stages, which was similar to the results of the correlation analysis between vegetation indices and nitrogen concentration of winter wheat plants in the study by Chen et al. [49]. Sensitive variables can effectively remove the redundant information of vegetation indices, reduce the complexity of the model, and thus improve the validation accuracy of the model. Some scholars have already conducted studies in this regard, such as Liu et al. [50], who constructed the model using seven kinds of vegetation indices after screening, which greatly improved the validation accuracy of the model.

### 4.2. Estimation of Nitrogen Content of Fruit Tree Canopy Leaves Based on Different Inversion Methods

In this study, three different inversion methods were used to estimate fruit tree leaf nitrogen content based on canopy multispectral data. Compared to the RF model, both the PLSR model and the RE model have lower model accuracy, as shown in Figures 9 and 10 of the revised version. This is because the RF algorithm is able to solve non-linear problems

and does not need to do any feature selection, the generalization error is estimated using unbiased estimation when creating the random forest algorithm, and the model generalization is strong. While the PLSR model and the RE model are both linear regression models, they are weak in terms of problem-solving ability. The three selected regression models, PLSR, RR, and RF, can be categorized as linear machine learning models and non-linear machine learning models, where RF is non-linear. Among the linear models, PLSR performed more prominently at all fertility stages, with predictive performance at all fertility stages of fruit trees, with RPDs greater than 1.4, whereas RR did not have predictive performance at the later stages of fruit expansion in 2022 (RPD < 1.4). The two models performed more similarly at other fertility stages, probably because RE, as a modified PLSR regression model [45,46], is more similar in its ability to deal with analytical problems, and both are able to deal with some multicollinearity problems, resulting in more similar results.

The RF model, on the other hand, showed strong ability. It had good performance in all four fertility periods of fruit trees, and their validation set $R^2$ was higher than 0.7 or more and RPD was greater than 2.0 or more at the late fruit expansion stage in 2022 as well as at the pre-fruit expansion stage in 2023, which provided excellent prediction ability. The study by Zha et al. [51] showed that when estimating the nitrogen nutrient index (NNI) of rice based on remote sensing by unmanned aerial vehicle (UAV), the RF model had the best accuracy, and the study by Osco et al. [52] reached the same conclusion. Of course, similar conclusions were found not only in estimating crop N content, N concentration, and NNI but also for other growth indicators. Wang et al. [53] showed that the use of the RF regression model can provide more accurate prediction accuracy for the prediction of leaf area index in rice, and the results of this study are consistent with this paper. Zheng et al. [24] estimated the nitrogen content of winter wheat leaves using multispectral UAV images combined with machine learning methods, and the results showed that the RF algorithm had the best performance. Barzin et al. [54] compared eight different machine learning methods to predict the nitrogen content of maize leaves using UAV images, and the results showed that the RF algorithm was one of the best-fitting models.

### 4.3. Insufficient Research

In this study, only the vegetation index was chosen as the independent variable to construct the prediction model of nitrogen content, which, in general, has some limitations to the improvement of model accuracy, while some studies have shown that the use of an approach based on the combination of the vegetation index and texture features can effectively improve the accuracy of the model, and the combination of the nitrogen content of plant leaves with the spectral information of the spectral data and the texture information of the image can improve the nitrogen prediction accuracy and generalization ability of the model [55,56]. The fusion of spectral and texture features is beneficial to alleviate the shortcomings of spectral analysis techniques with low sensitivity, thus improving the robustness of nitrogen prediction models [55]. Yan et al. [56] constructed a chlorophyll content prediction model based on the combination of spectral and texture features by comparing two algorithms, BPNN and SVM, and achieved better prediction results. Compared with other studies, this study has the problems of more influencing factors and a relatively small number of samples. Therefore, the subsequent optimization of the inversion model will consider different crops and increase the sample data to further improve the accuracy of the model application.

## 5. Conclusions

In this study, the multispectral UAV technology was used to estimate the total nitrogen content of apple trees in the flowering and fruiting stage, the pre-fruit expansion stage, and the post-fruit expansion stage in the southern border by utilizing linear parametric regression and machine learning regression. The results showed that machine learning regression was able to significantly improve the estimation accuracy of the full nitrogen

content of canopy leaves of fruit trees in South Xinjiang, especially the RF algorithm, which had higher accuracy in all three periods of fruit trees.

Therefore, it is estimated that the best choice for estimating the whole nitrogen content of fruit tree canopy leaves during the reproductive period based on multispectral UAV imagery is to use the machine learning method, and the RF algorithm can obtain the most prominent effect, which can provide a theoretical basis for the precise management of orchards.

**Author Contributions:** Y.G. and X.W. planned and designed the experiments; X.Z. and Z.Z. performed the experiments; X.Z. analyzed the data and wrote the draft manuscript; F.Z., Z.L. and J.L. helped with the experiment. All authors have read and agreed to the published version of the manuscript.

**Funding:** This study was supported by the Bingtuan Science and Technology Program (2022BC009), the Central Public-Interest Scientific Institution Basal Research Fund (IRI2023-19), and the Agricultural Science and Technology Innovation Program (ASTIP), Chinese Academy of Agricultural Sciences.

**Data Availability Statement:** Data are available upon request.

**Conflicts of Interest:** The authors declare no conflicts of interest.

## Abbreviations

| | |
|---|---|
| PLSR | partial least squares regression |
| RR | ridge regression |
| RFR | random forest regression |
| SVR | support vector regression |
| VARI | visible atmospheric impedance index |
| NRI | nitrogen reactivity index |
| SAVI | soil conditioning vegetation index |
| EVI | enhanced vegetation index |
| R-M | red edge model |
| NDRE | normalized red edge difference |
| GOSAVI | optimized green band soil-adjusted vegetation index |
| OSAVI | optimized soil-adjusted vegetation index |
| GBNDVI | normalized blue–green band difference vegetation index |
| NDVI | normalized difference vegetation index |
| RVI | ratio vegetation index |
| DVI | difference vegetation index |
| GNDVI | green band normalized vegetation index |
| TVI | triangle vegetation index |
| RMSE | root mean square error |
| RPD | relative prediction deviation |

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
