# Peer review of "An Estimation of the Leaf Nitrogen Content of Apple Tree Canopies Based on Multispectral Unmanned Aerial Vehicle Imagery and Machine Learning Methods"

_agronomy, doi:10.3390/agronomy14030552_

Round 1
Reviewer 1 Report
Comments and Suggestions for Authors
Review report about the “Estimation of Leaf Nitrogen Content of Apple Tree Canopy
based on UVA-multispectral Imagery and Machine Learning Methods” manuscript
The experimental topic is a current and important area. The importance and usefulness of data collected by remote sensing is constantly increasing.
The claims about the impact of high nitrogen use on the environment and on the food industry described in the introduction are correct, but globally, especially in Europe, nitrogen use is decreasing and therefore overuse is not a threatening problem. Mention should also be made of this.
The introduction is well prepared and contain a lot of new references from the last five years.
Materials and methoods
Please expand the agrotechnical description of the apple plantations used in the experiment in the material and methods section, as it is not very detailed.
Please add a characterisation of the total temperature and precipitation for the test period to the material and method section. Should create figures about it.
Please describe the reprojection errors of the orthomosaic. It should be below than 1 pixel.
The 50 meters flight altitude was calculeted from the ASL or AGL?
What was the angular position of the camera during the remote sensing?
What was the GSD of the completed orthomazik?
What time of day was the shoot and what were the cloud conditions like?
In table three the syllabification of the words is not correct please correct the formatting
Results
Please explain the different colours of the dots in Figures 8-10. Is it meaning the X and Y axis?
In the figure 8-10 the correlation between the measured and calculated value is not to high.
The manuscript contain high quality scientific results. Tho conclusion well preapred, and the authors write main results step by step, as a result, it can be interpreted well.
The results are adequate, but the presentation and comparison of the new results with the results of other international researchers requires a more detailed description.
Authors should make the suggested changes and complete the manuscript, after which it may be suitable for publication.
Reviewer 2 Report
Comments and Suggestions for Authors
The manuscript, entitled 'Estimation of Leaf Nitrogen Content of Apple Tree Canopy based on UVA-multispectral Imagery and Machine Learning Methods,' concerns the estimation of leaf nitrogen content of apple trees using three machine learning methods from drone-acquired images of an apple orchard. Three statistical methods were used to validate the quality of nitrogen estimation in the leaves of the studied trees: partial least squares regression (PLSR), ridge regression (RE), and random forest regression (RFR).
The manuscript is chaotically written and difficult to understand. The methods mentioned are statistical methods for evaluating expected results with obtained results, and I do not doubt this. The way the research process and the analysis of the results are presented in the manuscript could be more straightforward to understand. I have no comments on the Introduction chapter, whereas I have comments on the chapters:
Materials and Methods.
It is difficult for the reader to distinguish where the use of the trained model is described with the statistical methods used. According to the authors, is the use of typical statistical methods with a mathematical formula? The authors included the phrase "Machine Learning Methods" in the title, which is not synonymous with partial least squares regression (PLSR), ridge regression (RE), and random forest regression (RFR). The Machine learning process needs to be better described, while the statistical analyses are well described. Please explain in the text of the manuscript exactly how the different evaluations were carried out and complete the text with the neural network results.
Formula 3 - there needs to be an explanation of all the components of the formula in the text of the manuscript.
Technology roadmap - done carelessly, words cannot divide randomly, should be improved.
What is the text contained in lines 210 and 214? It contains a repetition of what is in Table 4; remove the text.
Results
Figures 4 - 7 are diagrams and are challenging to understand; they should be described in detail and explained in the text of the manuscript, especially how they are done. There is no indication in the figure what the blue circles mean, why some are dark and others light, why they are different sizes, what values are shown by the left vertical axis, and what values are shown by the lower horizontal axis. The very bad graphic quality of the drawings, some indicators on the drawing, difficult to read, drawings made carelessly. How do these diagrams relate to the individual items of the road map diagram? Describe and explain the drawings in detail in the manuscript's text and improve their graphic quality.
Figs: 8 and 9. What do the colors in these figures mean? Please explain where these values come from. Are they values resulting from calculations by partial least squares regression (PLSR), ridge regression (RE), and random forest regression (RFR) of individual indicators, or are they randomly selected nitrogen values? Does the neural network generate these results compared to the actual results, or are they just results obtained by using the vegetation indicators compared to the actual results? Also missing is a description of how this graph was created. There is a great deal of uncertainty here, requiring additional explanation. Please note that the reader should understand the text of the manuscript. The process of creating the graphs should be described in detail.
The text of the manuscript should be improved.
Reviewer 3 Report
Comments and Suggestions for Authors
The manuscript entitled Estimation of Leaf Nitrogen Content of Apple Tree Canopy Based on UVA-multispectral Imagery and Machine Learning Methodscan, in my opinion, be accepted after approval of the major revision
In my opinion, the topic of this paper is relevant to the journal Agronomy MDPI. The journal MDPI wants to see interesting and high quality papers.
Firstly, the paper has the following sections and subsections (i.e. Abstract, Introduction, Materials and Methods, Study area, UAV Imagery of apple tree canopy, Multispectral imaging and spectral data acquisition, Multispectral image, Total nitrogen content of apple leaves, Determination of vegetation index, Machine learning models, Model evaluation analysis, Technology Roadmap, Results, Relationships between the nitrogen content of the canopy leaves and vegetation index, Analysis of machine learning models for estimating the nitrogen content of apple leaves, Comparison of model accuracy, Discussion, Estimation of nitrogen content using vegetation index as independent variable, Estimation of nitrogen content of fruit tree canopy leaves based on different inversion methods, Insufficient research, Conclusion).
First, I highly recommend to the authors to include the abbreviations section before the conclusions section because this text contains many short terms and many specific scientific words.
The section Abstract
The Abstract section is generally well written, but at least one sentence should be added explaining the main findings of this research or how this research could be extended with new insights for future work. A sentence should also be added explaining how the Chinese Ministry of Agriculture has approved advanced systems in agriculture.
The section Introduction
Line 35, better explain a risk food security?
Line 55, how can UAV imagery improve remote sensing analyses in agricultural science. Is there any previous work or project in China that usually improved UAV imagery for apple tree canopy characteristics, better explain.
In this section, the authors need to better explain how the resolution of UAV imagery can contribute to better spatial analysis of apple trees.
The same needs to be used for machine learning approaches. The authors need to better explain previous machine learning approaches for the same or very similar studies in this paper.
Because of all these points, I strongly recommend the authors to read and cite two valuable research papers that explain better remote sensing and GIS techniques for estimating tree canopy properties. UAV imagery is not the only one that can be used to better estimate and analyse tree properties. Many other digital methods are also used. Two valuable works are:
- Valjarević, A., Djekić, T., Stevanović, V., Ivanović, R., & Jandziković, B. (2018). GIS numerical and remote sensing analyses of forest changes in the Toplica region for the period of 1953–2013. Applied geography, 92, 131-139. https://doi.org/10.1016/j.apgeog.2018.01.016.
- Zhou, X., & Kim, J. (2013). Social disparities in tree canopy and park accessibility: A case study of six cities in Illinois using GIS and remote sensing. Urban forestry & urban greening, 12(1), 88-97. https://doi.org/10.1016/j.ufug.2012.11.004.
The Materials and Methods
In Fig. 1, the geographical coordinates are difficult to recognise, so make them more visible.
Line 117, more about the accuracy of the sensors and the maximum height of the device?
Line 125 m, why this part started in bold is not necessary.
Line 145, how the authors divided nitrogen connect from the images made by drone?
There were some problems, for example dead pixels or turbulence of laminar layer in the air. Explain this better.
Line 156, I think that are more accurate determination of vegetation indices, so add some of them.
Model evaluation analysis
Does this recursive equation, because of the number of samples, explain better?
Technology roadmap
I do not think this subtitle is necessary
This subtitle (workflow) needs to be placed elsewhere in the text.
In this part of the manuscript, the authors need to write more about the meteorological conditions, especially the date of using the drone sample.
Results
How did the authors estimate the distribution of flowering. Did they use statistical software. Did the authors find the trend line. Explain this better.
Discussion
This section needs to be rewritten in some parts and the authors need to add more about previously published works and similar analyses.
Conclusion section
The authors must have answered the following questions.
Why is this research important?
How do the authors view this research in comparison to other similar research?
This work has the potential to be published. The authors have done a great job in this manuscript. The work is very interesting and scientifically correct.
In the end, I recommend a major revision.
Good luck to the authors
Reviewer #2

Comments on the Quality of English LanguageThe authors need to make small improvements to the English.
Round 2
Reviewer 2 Report
Comments and Suggestions for Authors
I accept the authors' comments and corrections to the text.
Reviewer 3 Report
Comments and Suggestions for Authors
The manuscript entitled Estimation of leaf nitrogen content of apple tree canopy based on UVA-multispectral imagery and machine learning methods be accepted in its present form.
The authors have responded to all my suggestions and recommendations. The manuscript now looks very good, with the concise and well-written text, with the very good maps and spatial representations.
Sincerely,
Reviewer#2